# Reproducing Visual Explanations of Variational Autoencoders

## Reproducibility Summary

**Scope of Reproducibility**

In this work we perform a replication study of the paper "Towards Visually Explaining Variational Autoencoders". This paper claims to have found a method to provide visual explanations of Variational Autoencoders (VAEs). The paper's primary claim is that their proposed method can generate gradient-based attention maps from the latent space of a VAEs. This is visually demonstrated on the MNIST dataset. Moreover, these attention maps are claimed to be useful for anomaly detection, which is demonstrated on the UCSD Ped1 and MVTec-AD datasets. Finally, this method is integrated into a loss function to obtain the attention disentanglement loss. This loss is shown to improve latent space disentanglement when integrated into a FactorVAE model, which is demonstrated on the dSprites dataset. This paper aims to reproduce all of the claims stated above.

**Methodology**

In order to produce attention maps which can localize anomalies for the MNIST dataset the code from the repository of the authors could be reused. Additional models were implemented for the UCSD Ped1 and MVTec-AD datasets based on supplementary material provided for the original paper. To reproduce the latent space disentanglement results, a AD-FactorVAE model – which combines the attention disentanglement loss and FactorVAE model – was implemented based on the original paper, the original paper's supplement, the FactorVAE paper and external code sources.

**Results**

The attention maps for the MNNIST dataset were successfully replicated. Nevertheless, we failed to reproduce the results for the UCSD Ped1 and MVTec-AD dataset. Furthermore, we were unable to reproduce the results for the AD-FactorVAE. Potential explanations for this could be the incorrect aggregation and/or weighting of the attention disentanglement loss, not training the models for enough epochs, or not using the correct method to produce the attention maps.

**What was easy**

Reproducing the attention maps and anomaly localization results for the MNIST dataset was relatively easy.

**What was difficult**

Reproducing the anomaly localization results for the UCSD Ped1, and MVTec-AD datasets was difficult. Additionally, reproducing the disentanglement results using the AD-FactorVAE proved to be problematic.

**Communication with original authors**

Except for a reply to our initial e-mail updating them of our attempt to reproducing the paper and asking if they would share more of their code, which they declined, the authors of the paper did not respond to any of our questions.

# 1 Introduction

This paper describes our attempt to reproduce the paper "Towards Visually Explaining Variational Autoencoders" by Liu et al [1]. Their paper introduces a method to provide visual explanations for variational autoencoders (VAEs) [2], which is inspired by the recent development of gradient-based attention maps for Convolutional Neural Networks (CNNs) that aid in visualizing and understanding these models [3]. The attention maps highlight areas of an image that are important for the classification (in the CNN case) or reconstruction (in the VAE case) of the respective image. It is important to search for these explanations since deep learning models tend to be difficult to interpret due to their complicated structure [4]. This complexity allows for superior performance but also prevents creators from understanding the reasons behind their models' outputs. If the models' underlying assumptions cannot be made explicit, there is a possibility that the predictions of the models rely on false assumptions. This could cause failure in later stages of deployment. Furthermore, adequate explanations for predictions can improve consumers' trust in the abilities and fairness of a model [5]. Additionally, we could learn new properties of the data with these explanations, and improve our own decision making abilities [6]. Deep models have been especially successful in the field of computer vision and have been widely adopted in real-life tasks which makes the demand for proper understanding of their performance all the more urgent [7, 8, 9]. The performance of techniques to produce explanations can generally only be determined by qualitative analysis, which is highly subjective and selective since not all train or test instances can be evaluated. Thus, we decided to reproduce a paper that not only promises to provide a method that can generate visual explanations for VAEs but also states that these attention maps produce state-of-the-art anomaly localization results and can be used to improve latent space disentanglement, which can both be quantitatively measured.

# 2 Scope of reproducibility

The original paper proposes a gradient-based technique to produce visual attention maps for VAEs [1]. It claims that these attention maps are not only useful as explanations of VAE predictions but can also be used to perform anomaly localization, which is demonstrated on the MNIST, UCSD Ped1 and MVTec-AD datasets, and latent space disentanglement, which is demonstrated on the dSprites dataset. In this paper we try to reproduce the attention maps and the results for these two practical use-cases. In order to determine whether the paper can be successfully reproduced, the following claims were constructed that capture the most important statements and which can be either confirmed by our research (in which case the reproduction was successful) or rejected (in which case our reproduction has failed).

- The method introduced in [1] is able to generate gradient-based attention maps from the latent space learned by a VAE model. These attention maps denote characteristics of a set of images that are important to the model and intuitively make sense, which is verified by qualitative analysis on the MNIST dataset.

- The attention maps are useful for anomaly localization, which is demonstrated on the MNIST data set, for which only qualitative analysis is provided. Additionally, it is shown on the UCSD Ped1 dataset where it obtains a better AUROC score than the *Vanilla-VAE* method that simply computes the difference between the original and reconstructed image. Finally, the performance for anomaly localization is state-of-the-art, which is shown on the MVTec-AD dataset where it obtains AUROC and IOU scores that are better or similar to those obtained by the models compared in the MVTec-AD paper.

- The attention maps can be used to improve latent space disentanglement by combining them into a loss function that can be added to the VAE loss during training. This is proven by incorporating the loss with the FactorVAE which results in a higher disentanglement score for the dSprites data set than the baseline FactorVAE model, while keeping a similar reconstruction loss.

# 3 Methodology

The following section discusses the implementation of the necessary functions to reproduce the claims stated in the previous section.

## 3.1 Attention maps

The first claim of the original paper is that the proposed gradient-based attention maps can explain the inner workings of a VAE model. Attention maps can be generated for each latent dimension separately. First, the image is forwarded through the model to obtain its latent representation. Subsequently, the activation of each of the dimensions of this representation is backpropagated to one of the last convolutional layers. The resulting gradients are aggregated per channel of the feature maps of the convolutional layer to obtain the weight given to each channel of the feature map.

This weighted sum of channels of the respective feature maps can then be used to retrieve the attention map of a single latent dimension. The method to obtain attention maps for anomaly detection is also closely related to this technique. The authors showcase how these attention maps explain the model by training models on a single class of images from the MNIST data set [10] and then showing attention maps generated by the model for images of other classes. These attention maps highlight areas which are not characteristic of the originally learned class.

The training of the VAE model was done by following the instructions in the repository with the code made available by the authors which included the main training procedure and the experimental VAE architecture. Small adjustments were implemented to make the core run on CPU as well. Acquiring the dataset, and transforming this to the right data structure was already done in the provided code as well. In addition to training the models ourselves, we had access to the pre-trained models which could be downloaded via a link in the authors' GitHub repository. The VAE structure – two convolutional layers followed by a fully connected layer in the encoder and a mirrored structure in the decoder – and the hyperparameters – image_size = 28 | batch_size = 128 | latent_space = 32 | learning_rate 0.001 – were described in the supplementary paper, which is available online[1], as well as provided in the code base. The amount of epochs was unclear, but we trained the model for 100 epochs since this was the default provided in the code.

The provided code already contained an implementation for attention map generation, which we could run after removing an erroneous parameter (gradient=one_hot) in the backpropagation function[2]. However, when consulting the paper it seemed that some parts of the code did not match the original method. By following the paper more closely we discovered that Liu et al. distinguish between two different methods, (1) the first is proposed to visualise explanations for VAEs by calculating the attention maps per latent dimension and aggregating these, and (2) the second is proposed specifically for anomaly detection where attention maps are generated from the sum of the inferred mean vector. In the provided code base only the (2) second method is implemented and so, in addition, we implemented the (1) first method ourselves.

Besides implementing both methods, we also chose to test various functions in place of the ReLU in function (2) of the paper. This was done as the provided code implemented the absolute value instead of the ReLU, which seemed to give similar results and we were curious to see whether the sigmoid could be used as well, since this is also a common function used to scale values between 0 and 1. To conclude, we tested both the pre-trained models and the models trained from scratch with the two different methods and the three different functions.

## 3.2 Anomaly detection

The second claim by Liu et al. is that the attention maps are not only useful for anomaly localization but can produce state-of-the-art results. They show this by conducting qualitative and quantitative experiments on two different datasets for anomaly detection: the UCSD Ped1 dataset [11] and the MVTec-AD dataset [12].

### 3.2.1 UCSD Ped1 set up

The UCSD Ped1 dataset contains video frames of a pedestrian walkway where all non-pedestrains are considered to be anomalies. The implementation for the anomaly detection on the UCSD Ped1 dataset was not provided in the authors' code. For the qualitative research we could use the same approach as in the previous section, namely training a VAE model on the UCSD Ped1 training set – which does not contain any anomalies – and afterwards generate attention maps of the test set – which does contain anomalies – for each of the convolutional layers of the VAE using the (2) second method. The changes we had to make for these experiments consisted of implementing a different architecture for the VAE model containing three convolutional layers instead of two and adding functions to properly read in the UCSD Ped1 training and test set, which included resizing the images to 100 by 100 pixels. Table 2 in the addendum paper shows the architecture of the VAE model that was used for these experiments and specifies a learning rate of 0.0001 with a batch size of 32 frames for training. The amount of epochs was unclear, but we ended up training the model for 1000 epochs, which took around 4 hours on one NVIDIA 1080 GPU.

For replicating the quantitative results, we implemented functions to evaluate the produced attention maps. The UCSD Ped1 test set contains both video frames containing anomalies as well as ground truth masks denoting the location of a possible anomaly. Given the anomaly attention maps, binary anomaly localization maps were generated using a variety of thresholds whose respective overlap with these ground truth masks is encapsulated in a pixel-level true positive rate (TPR), false positive rate (FPR) and ROC curve. Subsequently, these could be used to calculate the area under the ROC curve (ROC AUC) score using the sklearn metric 'roc_auc_score'. Furthermore, with the trained model we tried to reproduce the baseline score presented in the paper, which involved computing the difference between the input image and its reconstruction and similarly comparing the result with the ground truth masks to obtain the ROC AUC score.

---

[1] https://openaccess.thecvf.com/content_CVPR_2020/supplemental/Liu_Towards_Visually_Explaining_CVPR_2020_supplemental.pdf

[2] The original authors have recently also updated their implementation in the same way we did.

### 3.2.2 MVTec-AD set up

In the paper, the anomaly detection is further tested on the in 2019 released MVTec Anomaly Detection (MVTec-AD) dataset [12]. This dataset consists of images of 15 natural objects and textures with different defects and pixel-level ground truth masks. Again the repository did not contain any code to run experiments specifically for this dataset, but the functions to obtain the attention maps using method (2) could be used. Additionally, we implemented the VAE as stated in Table 3 of the addendum paper by the authors. The encoder consists of a ResNet18 (except for its last two layers) followed by two linear layers and the decoder consists of two linear layers, six blocks containing a 2D Convolutional layer, batch normalization and a ReLU, and finally a sigmoid layer. During training, Adam was used for optimization with a learning rate of $1e-4$ and batch size of 8. We trained a separate model for each object/texture category. Again, the number of epochs was undeclared, but we trained each category for 100 epochs due to time constraints.

Before training, the data was augmented similarly to the MVTec-AD paper, of which the specifications are as follows. All images were resized to $256 \times 256$ pixels. Additionally, they were randomly rotated between [-30°, +30°] and/or horizontally mirrored to create an augmented training set of 10.000 images per object/texture. The probability for the horizontal mirroring was not specified but chosen to be 0.5 because it is the default value for this method. The padding used after rotation was also unclear. We therefore decided to use black pixels since that is the standard parameter for padding. For comparison, the *leather* category was padded differently, namely with the colour of the original image's upper-left pixel.

After training, the attention maps for the images containing anomalies could be generated for the last convolutional layer of the VAE model. These attention maps could then be compared to the ground truth masks to obtain the ROC AUC score. Moreover, the intersection-over-union (IOU) score was also calculated from the ROC curve.

The experiments were run on the same GPU node as the USCD data set. Creation of the augmented dataset took around 1,5 hours per object, while training of one model for 100 epochs took around 2,5 hours. The MVTec-AD dataset has a size of 5GB and training requires around 11GB of memory

### 3.3 Latent space disentanglement

The authors' code did not contain any implementation specific to the proposed attention disentanglement (AD) loss. Nonetheless, the code to generate attention maps applying the (2) method proved to be useful. Consequently, we had to implement the FactorVAE, disentanglement metric, and calculation of the AD loss ourselves. The authors use the FactorVAE [13] to showcase how the proposed loss can be integrated into other loss terms. They call this new version of the model AD-FactorVAE. To implement FactorVAE, we adapted an external GitHub repository [14] which was recommended to us by the original authors. From this repository we also adopted code to read in the dSprites dataset, which is used for the evaluation of both the FactorVAE and AD-FactorVAE [15]. We integrated the proposed loss into this implementation according to Equation 6 in the paper. Besides the model, the paper also makes use of the disentanglement metric proposed in the FactorVAE paper. The underlying assumption of this metric is that if the latent space is perfectly disentangled, then each latent dimension of the representation should correspond to a single latent factor used to generate the data. Consequently, if that latent factor is kept fixed, the variance of the corresponding latent dimensions should be 0. A majority vote classifier is trained to predict factors corresponding to specific latent dimensions by fixing single factors, passing data through the trained VAE model and checking for the dimension with the lowest variance. The metric is the accuracy of this classifier. The implementation of this metric was adopted from an external GitHub repository [16].

The hyperparameters used for these experiments are shown in Table 1. The original paper states that they did not modify any parameters of the model compared the ones mentioned in the FactorVAE paper [13], other then increasing the number of latent dimensions from 10 to 32. The negative slope of the LeakyReLU in the Discriminator is not mentioned in the FactorVAE paper, however the adapted implementation assumes it to be 0.2. Moreover the authors also did not specify the size of the test set used to obtain the disentanglement score. We chose to use 200 votes, because this did not slow down the score calculation too much while still providing us with a robust indication of the accuracy. Additionally, some hyperparameters of AD-FactorVAE were unclear from the paper. There is no optimal value stated for $\lambda$, the weight of the AD loss. It was also not clear how to aggregate AD losses of several latent dimension pairs or how to pick one or more pairs for the AD loss in general. However due to time constraints, we could only experiment with the different aggregation of AD losses for all possible dimension pairings. The experiments were again run on one NVIDIA 1080 GPU node. Training requires around 16GB memory and 24 hours to finish, while testing requires the same amount of memory and at most 30 minutes.

| Optimizer: | Adam | Learning rate: | $10^{-4}$ | Batch size: | 64 |
|---|---|---|---|---|---|
| Latent dimensions: | 32 | VAE beta1: | 0.9 | VAE beta2: | 0.999 |
| **LeakyReLU slope**: | 0.2 | Discriminator beta1: | 0.5 | Discriminator beta2: | 0.9 |
| Metric batch size: | 100 | Metric training size: | 800 | **Metric evaluation size**: | 200 |
| $\gamma$ value: | $\{20, 40, 100\}$ | **$\lambda$ value**: | 1 | **Aggregation method**: | {sum, mean} |

Table 1: Hyperparameters used for AD-FactorVAE. **Bold** parameters are not clearly defined in the paper.

Figure 1: Qualitative results on the MNIST dataset with activation functions along the y axis and the various methods along the x axis. The algorithms were trained on handwritten 1's and per method three random 7's are visualised.

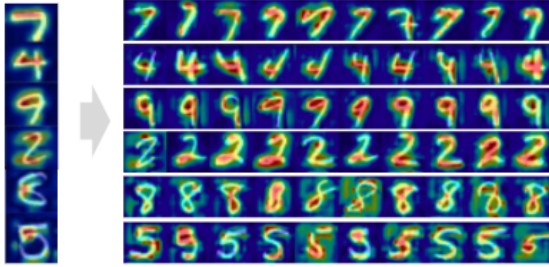

Figure 2: Qualitative results on the MNIST dataset. On the left are the original results and on the right are randomly sampled results produced with method (1) and ReLU.

## 4 Results

This section states the results of the experiments described in the previous section.

### 4.1 Attention maps

Replicating the results for the MNIST dataset was an involved process – as described in the methodology – leaving us with various versions of replications. In Figure 1 we present the two different methods based on the pre-trained model trained on handwritten digit 1's and tested on handwritten digit 7's with three different functions (ReLU, absolute value and sigmoid). Most of these versions would be able to produce the images presented in the paper, except for those generated using the sigmoid which is why this function was excluded from further experiments. The only difference between the decent versions is the level of cherry picking Liu et al did. We are assuming a minimal amount, and therefore conclude that the examples in Figure 4 in the original paper were most likely produced with method (1) and the ReLU function, even though the paper seems to suggest they were produced using method (2). We chose to proceed with method (1) as the results resembled those presented in the paper much better. The attention maps for the model trained on digit 1 and tested on the digits 9, 2 and 4, as well as the model trained on digit 3 and tested on 8 and 5 are produced with those settings and the pre-trained models. These can be found in Figure 2.

We are fairly certain that the differences between the pre-trained model and the models trained from scratch are not due to differences in architecture or hyperparameters as they were explicitly stated in the addendum paper, which means the only remaining influential factor is the amount of epochs for which the model was trained. As we reached out to the authors about the amount of epochs but did not get a response we could not confirm this suspicion.

### 4.2 UCSD Ped1 Results

We were able to generate qualitative results which match those presented in the paper to a certain degree, however, they are not as good in quality. These results can be seen on Figure 3. The frames in the paper could be matched in quality if we were cherry picking results, which could be one of the explanations for the mismatch, but a more likely explanation would be that we were simply not able to replicate the results properly.

Quantitative results are shown in Table 4. These results show that we were unable to recreate the exact scores presented by Liu et al. It is worth noting that we were also not able to exactly reproduce the baseline scores presented by Liu et al. for the VAE model (also *Vanilla-VAE*). The baseline score in the paper reached 0.86 while ours got stuck at 0.82. In conclusion, all our ROC AUC scores were significantly lower compared to the scores from the paper. The scores were

highly dependent on the method, and while method (2) was presented as the anomaly localisation method, method (1) seems to consistently perform better.

An additional topic of uncertainty was how masks without anomalies were handled. As it was not touched upon in the paper or the addendum paper we explored two options, (1) including all images whether they had anomalies or not and (2) excluding those frames without any anomalies. Furthermore, because we created this code from scratch we experimented with various small adjustments to the code, one of which was whether we should upsample or downscale the images, depending on the convolutional layer. Where the second (downscaling) led to higher scores, we decided to stick with the first approach as we felt it was more representative of the actual performance of the model (downscaling increases the probability of getting the right pixel value, so as we apply more convolutions the image gets smaller which leads to higher scores). Another question left unanswered was whether to do any processing on the generated attention maps, such as normalization etc. We decided against this, as it did not improve the scores and the normalization in the code base seemed to only have the goal of displaying the images correctly.

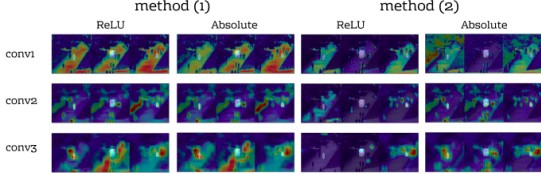

Figure 3: Attention maps for anomaly detection on UCSD, for different layers and generation methods.

| | Paper | (1) relu | (1) abs | (2) relu | (2) abs |
|---|---|---|---|---|---|
| baseline | 0.86 | 0.82 | 0.82 | 0.82 | 0.82 |
| conv1 | 0.89 | 0.71 | 0.70 | 0.56 | 0.59 |
| conv2 | 0.92 | 0.80 | 0.78 | 0.63 | 0.69 |
| conv3 | 0.91 | 0.68 | 0.73 | 0.52 | 0.64 |

Figure 4: AUROC scores on UCSD

## 4.3 MVTec-AD Results

The MVTec-AD results are shown in Table 2. Due to time constraints we were not able to produce scores for all objects and textures and could only use the (2) second method for attention map generation. We see that our scores are significantly lower compared to the scores stated in the paper and presented in the second column of Table 2. The results are also lower compared the scores presented in the original work of Bergmann et al. [12] which are presented in Table 2 of the original paper by Liu et al. Figure 5a shows some qualitative results from three categories. These attention maps were one of the best that were produced during our experiments. One can note that these attention maps are far from accurate in selecting the anomalous regions. Therefore both the quantitative and the qualitative results we produced are quite different from the results in the original paper. However, in contrast to the UCSD Ped1 experiments, there is no baseline score we can use to put the scores into perspective.

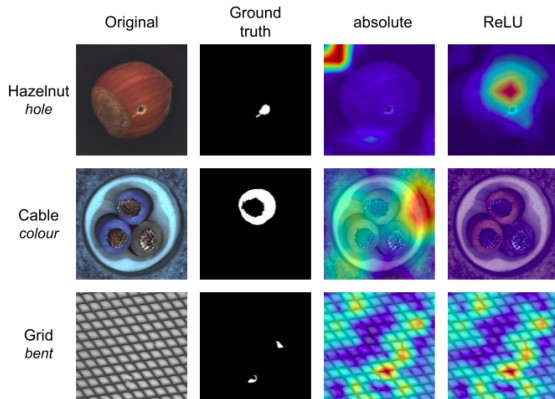

(a) Qualitative results for Hazelnut, Cable and Grid categories in MVTec-AD. For each category, we show the original image, the ground-truth mask, the attention maps generated with ReLU and with absolute value

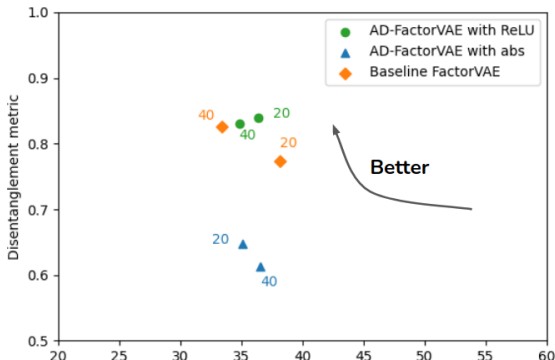

(b) Achieved disentanglement scores of FactorVAE and AD-FactorVAE models. The numbers next to the markers indicate $\gamma$ parameter values.

Figure 5

## 4.4 Anomaly detection

In conclusion, we can state that we were not able to replicate the results for anomaly detection proposed by the authors. We therefore cannot support the claim that the performance of their proposed anomaly localization method is state-of-the-art by obtaining ROCAUC and IOU scores that are better or similar to those obtained by the models compared in the original work of Bergmann et al. Moreover, the attention maps did not exceed the baseline score for the UCSD Ped1 experiments.

There might, however, be room for improvement within our implementations which could lead to state-of-the-art performance, as stated by Liu et al. Possibilities for improvement could be training the models for more epochs, different weight initialization, or changing the way we generate the attention maps, for which there is some confusion about the usage of method (1) or method (2), and using ReLU or absolute value.

Striking is the fact that method (1) – which was originally just proposed for visualising the attention maps – consistently outperforms method (2) – which was specifically proposed for anomaly localization – on the anomaly localization task and is able to generate results much closer related to those presented in the paper. The trade-off between the two functions seems to be quality vs. time. Method (1) has to loop over the entire latent space, making it much slower, where method (2) just has to propagate the mean vector. We are unsure which one of these methods was actually used to obtain the final results presented in the paper.

## 4.5 Attention Disentanglement Results

As the FactorVAE paper does not mention the weight initialisation used, we relied on the external code which provides options for both normal and kaiming normal initialization. However after being unable to reproduce the baseline results with either of these methods, explicit weight initialization was removed causing the pytorch default, kaiming uniform, to be applied. This allowed us to achieve the desired results, but it also shows how sensitive these methods are to small technical details. Additionally, we had to ignore the color factor to reach the correct disentanglement scores. Object or background color cannot be varied randomly in the dSprites dataset hence it always acts as a fixed factor for the metric.

Setups where loss values were aggregated using the mean and a $\gamma$ value of 20 or 40, converged successfully to the expected reconstruction loss. Results of their corresponding scores can be seen on Figure 5b. We can clearly see that using ReLU achieves significantly superior results compared to using absolute value. However, in general we can see that adding the proposed loss did not increase the disentanglement score compared to the baseline FactorVAE.

| Category | theirs | ours ReLU | ours abs |
|---|---|---|---|
| Carpet | 0.78 0.1 | 0.51 0.02 | 0.47 0.02 |
| Grid | 0.73 0.02 | 0.55 0.02 | 0.50 0.02 |
| Leather (diff. padding) | 0.95 0.24 | 0.52 0.13 | 0.67 0.18 |
| Tile | 0.80 0.23 | 0.59 0.14 | 0.62 0.14 |
| Wood | 0.77 0.14 | 0.42 0.05 | 0.63 0.07 |
| Bottle | 0.87 0.27 | 0.53 0.08 | 0.46 0.08 |
| Cable | 0.90 0.18 | 0.50 0.05 | 0.62 0.08 |
| Capsule | 0.74 0.11 | 0.54 0.03 | 0.69 0.04 |
| Hazelnut | 0.98 0.44 | 0.63 0.08 | 0.75 0.09 |
| Metal Nut | 0.94 0.49 | 0.39 0.15 | 0.46 0.15 |
| Pill | 0.83 0.18 | 0.53 0.05 | 0.60 0.06 |
| Screw | 0.97 0.17 | 0.52 0.00 | 0.51 0.00 |
| Toothbrush | 0.94 0.14 | 0.64 0.04 | 0.65 0.03 |
| Transistor | 0.93 0.30 | 0.47 0.12 | 0.61 0.15 |

Table 2: Quantitative results for pixel level segmentation on 14 categories from MVTec-AD dataset. For each category, we report the area under ROC AUC curve on the top row, and best IOU on the bottom row.

Setups where the loss values were aggregated using a sum or a $\gamma$ of 100, converged to reconstruction loss values well over 100. A possible reason to why summing does not work is that then the AD loss gets too large and influences the overall loss too much. The same could apply to the issues with a large $\gamma$ parameter and the Total Correlation loss of FactorVAE, however as this was part of the baseline, this rather suggests that we might have missed key information about training models with a parameter of this magnitude. Because the authors claim the addition of the AD loss does not decrease the reconstruction loss, we only determined a model's disentanglement score if it reached a reconstruction loss similar to the baseline FactorVAE model.

## 5 Discussion

In this paper, we have provided insights in the reproducibility attempt of the paper "Towards Visually Explaining Variational Autoencoders" by Liu et al. Our results show that, with the provided tools and within a time-frame of four weeks, we have been able to reproduce similar attention maps for the MNIST dataset. The gradient-based attention

maps clearly denoted characteristics that intuitively made sense. We can therefore confirm the first claim by the authors. Besides that, we have not been able to produce any results that were comparable to the ones from paper. This leads to the conclusion that we cannot successfully confirm the second and third claim made by the authors. However, some results and extra experiments indicated that there might be room for improvement within the used methods through hyperparameter tuning or small model changes. Our approach has several weaknesses which will be discussed in the next section.

## 5.1 Strengths and weaknesses

Our implementation has several strong points, including:

- The authors' code was partly available on GitHub containing method (2) for the generation of the attention maps (as explained in Section 3.1) as well as the set-up for the MNIST experiments. This means that for these parts, we are quite certain that our code matches the authors' code.

- All datasets were publicly available and well documented, causing us to be fairly confident that we used the same data as the authors.

- For the models that were not implemented in the authors' code, the architectural details were publicly available in the authors' addendum paper. We were therefore able to implement the same models as the authors used.

- For the FactorVAE model, the authors referred us to a repository which contained an implementation that only needed some small adjustments to give us the correct baseline results. Thus, it seems unlikely that our implementation of the FactorVAE model is incorrect.

- For the FactorVAE disentanglement metric a Google repository was found that contained a proper implementation, which gives a high probability that our implementation of this metric is correct.

- In many of our experiments, we have tried several different approaches to obtain the same results as the authors. For example, because we did not obtain correct results for the MNIST experiments, we also implemented the first approach for the attention maps generation even though the paper mentioned the results were obtained using the second approach. For the attention disentanglement we experimented with several attention map generation and loss aggregation methods.

However, there are also some weaknesses in our approach:

- The authors' code and the paper were sometimes inconsistent with each other. Especially for the generation of the attention maps. It was unclear if the results were obtained using ReLU (as stated in the paper) or using absolute value (as stated in the code). We tried to overcome this weakness by implementing both approaches. Furthermore, it was unclear which method to use for which part. The misalignment between the fact that method (2) seems to be presented as the method to use for all experiments in the original paper, while method (1) consistently brings us results that are closer to the results presented is vexing to us and might account for some of the discrepancies between the original results and ours.

- For each experiment the amount of training epochs was unclear causing us to make guesses due to limited time.

- It showed to be very difficult to reproduce the qualitative and quantitative results for anomaly localization on both the UCSD Ped1 dataset and the MVTec-AD data even though the models' architectures were specified in the addendum paper. Additional research is needed to discover why our experiments failed to produce the correct scores.

- It was not possible to implement the exact same data augmentation for the MVTec-AD dataset as the paper did not provide enough information on this topic.

- The entire implementation for the attention disentanglement was missing in the authors' code. We therefore had to make many guesses about the implementation of this part, which made it difficult to reproduce the qualitative and quantitative results for latent space disentanglement on the dSprites dataset.

## 5.2 Communication with original authors

We have communicated with the authors on two occasions. The first time we e-mailed them with a request for the complete code as the public repository had many missing parts. They replied that they were not able to give us their full code as they were planning to patent their implementation in the future. We sent them another e-mail with some follow-up questions about the number of epochs, data-augmentation for the MVTec-AD dataset and method of aggregation of the AD loss. Unfortunately, we did not receive a reply to that last e-mail.

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
