# OpenReview forum: "Reproducing Visual Explanations of Variational Autoencoders"
_ML_Reproducibility_Challenge/2020 — Reject_

### Official Review · AnonReviewer1 · 2021-02-11

**Rating:** 4
**Confidence:** 4

**Review:**

Authors attempted to reproduce the main experiments in the original paper, namely (1) per-image attention maps on MNIST, (2) anomaly detection explanations on MNIST, (3) anomaly detection on UCSD Ped1 and MVTec-AD, and (4) latent space disentanglement on dSprites. The authors of the original paper only provided code for generating visual explanations on MNIST.

Authors distinguish two visualization methods: (1) visualizing explanations for VAEs by calculating the attention maps per latent dimension and aggregating these, and (2) generating attention maps from the sum of the inferred mean vector in order to visualize anomalies. Authors then compare the two methods for anomaly detection, which I believe is incorrect. (1) aims at highlighting the areas of the input image that explain the data (e.g. if the VAE was trained on 1's, which regions of the image look like a 1?). On the other hand, (2) aims at highlighting the regions of the input image that are not well explained by the distribution of the training data (i.e. if the VAE was trained on 1's, which regions of the image do *not* look like a 1?). Therefore, the first method should be used to reproduce Figure 1 in the original paper, whereas the second method should be used to reproduce Figure 4.

Authors used the provided code for the aforementioned experiments, where they had to remove an input parameter to make it work. There is an issue on this regard in the repository, where the first author of the original paper wrote the following: "[...] To me, it definitely has to do with the new updates of PyTorch. But unfortunately I'm not exactly sure how they affect auto-gradients in our case.". I would encourage authors to double-check that the code behaves as expected, or run the provided code using version 1.0 of PyTorch as suggested in the repository.

The remainder of the submission attempts to reproduce the rest of experiments, but all results (including baselines) are quite far from those reported in the original paper. It is unclear why this happens and, while authors suggest that it might be due to shorter training runs, I suspect that this might also be related to some mistakes in the implementation of the attention maps.

In general, the submission failed at reproducing the original results. It is unclear whether this is due to a difference in the experimental setup or due to implementation errors. For this reason, I lean towards rejecting the paper and encourage authors to investigate why the results on MNIST could not be replicated before moving on to the more complex experiments in the paper.

Minor comment: captions for Figures 1 and 2 are very short and it is difficult to understand what is being shown.

Some typos:
- Reproducibility Summary: papers -> paper's, its' -> its
- Introduction: models -> model's (or models')
- Figure 2: Repliction -> Replication
- UCSD Ped1 Results: preform -> perform

**Familiar With The Original Paper:**

I have read the original paper

**Reproducibility Summary:**

Report has summary

---

### Official Review · AnonReviewer2 · 2021-02-26
**Reproducibility report on Visual Explanations of Variational Autoencoders**

**Rating:** 9
**Confidence:** 4

**Review:**

This review is detailed and high quality. It aims to reproduce VAE results on multiple data sets.

The reproducibility report is robust since the authors are able to replicate results in some of the data sets (MNIST). This, together with the detailed reporting, demonstrates that they had acquired the necessary understanding of the methodology in order to reproduce the original analyses. Reproducibility efforts fail in other data sets, and plausible explanations are provided. Communication with the original authors has taken place appropriately.

The work is significant as it robustly highlights potential problems in the original publication in terms of reproducibility and includes thorough reporting and discussion about the contributing factors.


**Familiar With The Original Paper:**

I have not read the original paper

**Reproducibility Summary:**

Report has summary

---

### Official Review · AnonReviewer3 · 2021-03-02
**In this paper, it discussed the setup for the experiments in detail and explained the difficulties they were facing to reproduce the results.**

**Rating:** 5
**Confidence:** 2

**Review:**

This paper provides a detailed description of the setup for reproducibility and well-discussed experiments and results. Given the limited resources and lack of support from authors of the original paper, it shows the difficulties for reproducing the main results for anomaly detection.

It is not very clear about the scope of reproducibility based on the description from the reproducibility summary on page 1. It sounds more like a description of the main results in the original paper.

“To produce attention maps to localize anomalies for the MNIST dataset the repository of the authors could be used.” It is not clear here whether reproduced from the reused author repository or not.


The writing of this paper needs to be polished. Examples below show the sentences that may need to rephrase:
“The papers primary claim is that ...”
“ ..., its’ supplement and external source code.”


**Familiar With The Original Paper:**

I have not read the original paper

**Reproducibility Summary:**

Report has summary

---

### Decision · Program_Chairs · 2021-03-31

**Decision:**

Reject

**Comment:**

Overall reviews and/or the paper content not good enough for the AC to recommend to the journal.